# OpenReview forum: "Multi-Agent Causal Discovery Using Large Language Models"
_ICLR.cc/2025/Conference — ICLR 2025 Conference Withdrawn Submission_

### Official Review · Reviewer_mWGv · 2024-11-03

**Soundness:** 3
**Presentation:** 2
**Contribution:** 2
**Rating:** 5
**Confidence:** 3

**Summary:**

This paper proposes a multi-LLM agent framework for causal discovery. The framework, particularly the Hybrid Model, leverages cooperation and debate among different LLM agents, along with their function-calling capabilities to employ statistical algorithms for causal discovery. The designed experiments demonstrate promising results by integrating LLMs' internal knowledge, agent debate and cooperation, and statistical algorithms. This framework lays a foundation for future research on multi-agent LLMs in causal problems.

**Strengths:**

Strengths:
- The authors aim to establish a multi-LLM agent framework for addressing causal discovery problems, leveraging advanced statistical algorithms and the internal knowledge of LLMs to achieve more accurate discovery results, as LLMs alone may not produce optimal outcomes in causal discovery.
- The authors conduct a full graph discovery experiment using the latest LLM models.
- The Meta-Debate module is designed to enhance the reasoning capabilities of LLMs.

**Weaknesses:**

Weaknesses:
- There are some formatting errors, such as unresolved references like "Figure ?."
- Given that the causal discovery results provided by LLMs alone sometimes lack stability [Altering the order of variables in the prompts can impact the results], presenting statistical significance results and trying more data source could enhance the validity of the findings.
- The experiments might not fully capture the complexity of real-world datasets. The generalizability of the model to more diverse settings remains unclear.

**Questions:**

Questions:
- Given that prompt-based approaches for LLMs can sometimes yield unstable results, would you consider including statistical significance results to enhance robustness?
- Conducting full graph discovery on three datasets may be insufficient to fully demonstrate the framework's validity. Would you consider including additional data sources and exploring more causal settings, such as causal event identification?
- Here are some relevant works that could strengthen the validity of your findings:

   [1] Improving factuality and reasoning in language models through multiagent debate. ICML 2024

   [2] Causal Parrots: Large Language Models May Talk Causality But Are Not Causal. TMLR 2023

   [3] LLM4Causal: Democratized Causal Tools for Everyone via Large Language Model. COLM 2024

   [4] Causal evaluation of language models.

   [5] Is Knowledge All Large Language Models Needed for Causal Reasoning?

---

### Official Review · Reviewer_UzsQ · 2024-11-03

**Soundness:** 3
**Presentation:** 2
**Contribution:** 2
**Rating:** 3
**Confidence:** 4

**Summary:**

This paper introduces a MAC for causal discovery using LLMs. The authors propose three distinct models: the Meta Agents Model, the Coding Agents Model, and the Hybrid Model. The Meta Agents Model uses multiple rounds of debate among different agents to discuss and identify causal relationships. The Coding Agents Model combines statistical causal algorithms, employing code writing and execution to identify causal links. Finally, the Hybrid Model merges the strengths of the previous two models, using both multi-agent statistical analysis and logical reasoning to construct causal graphs.

**Strengths:**

The use of cooperation and competition among multiple agents enhances the accuracy and comprehensiveness of causal discovery.

The framework combines the characteristics of different models for more efficient and flexible causal analysis.

The paper demonstrates the performance of different models on various datasets, using metrics such as SHD and NHD for comparison.

Exploration of LLM Potential in Causal Inference: This is the first in-depth study of LLMs’ multi-agent approaches in causal discovery, highlighting the potential for applying this method across diverse fields.

**Weaknesses:**

A hybrid model can balance statistical inference with the reasoning abilities of a language model, making it suitable for tasks with high complexity. However, the hybrid approach does not consistently outperform other methods significantly.

While multi-agent systems can enhance model performance, their effectiveness largely depends on the complexity and structure of the dataset. For example, Coding Agents perform well on moderately complex datasets, but their performance may fall short for simpler or highly complex cases. This suggests that models may require tuning or selection based on the dataset, which increases the usage barrier.

Due to the iterative debugging needed in Coding Agents models during code execution, handling longer or more complex code may lead to errors or debugging loops. Furthermore, coordinating multiple agents can, in some cases, produce unexpected outputs, requiring additional debugging and testing to ensure stability.

When processing larger datasets, multi-agent models—particularly the Coding-Debating Hybrid and Debating-Coding Hybrid models—consume a significant amount of tokens. Although the framework incorporates methods to reduce token consumption (such as limiting debate rounds), token usage remains a critical constraint for applications with limited resources.

The paper also has numerous issues in expression. For instance, the citation format in the introduction and related work sections needs improvement, and lines 463 and 478 lack references to specific images.

**Questions:**

See the weaknesses

---

### Official Review · Reviewer_XQTC · 2024-11-03

**Soundness:** 3
**Presentation:** 3
**Contribution:** 2
**Rating:** 5
**Confidence:** 3

**Summary:**

Driven by the potential of multi-agent large language models (LLMs) in complex problem-solving, the authors propose agent-based frameworks to enhance causal discovery. They introduce three multi-agent frameworks (models): (i) the Meta-Agent Model, featuring two debate agents and a judge agent who collaboratively perform causal discovery through reasoning and discussions; (ii) the Coding Agent Model, which operates in two phases: first, two debate agents and a judge agent determine the appropriate causal discovery method and devise an execution plan; in the second phase, they write and run the code using advanced statistical libraries; and (iii) the Hybrid Model, which integrates both the Meta-Agent and Coding Agent models. Experiments on three datasets demonstrate that these frameworks outperform baseline models, including traditional causal discovery algorithms and standard LLMs.

**Strengths:**

- It is the first to leverage an LLM multi-agent system for causal discovery.
- The paper proposes a multi-agent framework for causality, presenting three distinct models that demonstrate strong performance in experiments on real datasets.
- It provides an extensive comparison of diverse LLM models and includes an informative ablation analysis to illustrate the necessity of each key component in the proposed method.

**Weaknesses:**

- The paper’s methodological novelty may be somewhat limited, as its primary contribution appears to be the use of a multi-agent debating mechanism to improve the precision of causal reasoning, with much of the effort focused on constructing prompts tailored to causal discovery-specific problems. It would be valuable if the authors could further elaborate on any additional innovative aspects of the design.
- Clarity could be improved throughout the paper. For example, i) the distinction between the debate-coding module and the coding agent is unclear; ii) some figure references are missing, iii) there is a lack of explanation of rationale behind the hybrid models, such as how information from the coding agent enhances the performance of the meta-agent and how prior knowledge from the meta-agent is utilized within the coding agent.
- The experimental results may not be sufficient to fully demonstrate the method’s robustness, as the performance comparison is limited to only three datasets, which might not provide a representative evaluation. Additionally, including a more detailed discussion on the stability of LLM responses could be helpful.

**Questions:**

- In Figure 2, the prompt in the role section concludes with 'Today is <current date>.' Is this addition essential for performance?
- How does the debating phase in the coding agents model determine the appropriate algorithm? Model selection can be challenging, even for domain experts, as algorithms often rely on different assumptions about the data, which are difficult to verify. Does the proposed method account for this when recommending an algorithm?
- In the debate-coding module, the prompts enumerate all causal discovery methods under consideration, detailing their functions and parameters as sourced from a Python library. Will there be any potential scalability challenges, if more methods are to be included? Additionally, might the accuracy of method recommendations be impacted by the growing number of methods included in the prompts?
- In the debate-coding module visualization in Appendix A2.2, several additional steps are included in the final suggested plan beyond implementing a causal discovery algorithm, such as interpretation, model evaluation, .etc. How are these other steps executed?
- In the experimental results, the coding agent outperforms traditional causal discovery methods on the Sachs dataset. Could you explain the rationale behind this improvement? From my understanding, the key difference introduced by the coding agent, compared to traditional causal discovery algorithms, is the debating phase for method selection and planning. Intuitively, shouldn't the performance of the coding agent align with the direct implementation of the algorithm chosen during the debating phase?
- In the ablation analysis section, 'Single-Agent vs. Causal Agent Debate,' what prompt is used for the single agent? If we instruct a single LLM to handle both the debating and judging processes during reasoning, would its performance be comparable to that of the multi-agent framework?
- In the ablation analysis section, 'Eliminating the Judge,' why not include a comparison with GPT-4o mini?

---

### Official Review · Reviewer_d8LD · 2024-11-03

**Soundness:** 2
**Presentation:** 3
**Contribution:** 1
**Rating:** 3
**Confidence:** 4

**Summary:**

This paper explores the use of multi-agent Large Language Models (LLMs) for causal discovery, introducing a new framework called Multi-Agent Causality (MAC). MAC includes three distinct models—Meta Agents Model, Coding Agents Model, and Hybrid Model—each designed to leverage multi-agent interactions to enhance causal inference. The framework addresses several key challenges in causal discovery, such as harnessing agents for debate (Meta Agents Model), employing coding agents for data-driven causal graph construction (Coding Agents Model), and combining both approaches to optimize performance (Hybrid Model). The study assesses the efficacy of MAC across three datasets (Auto MPG, DWD Climate, and Sachs Protein) using metrics like Structural Hamming Distance (SHD) and Normalized Hamming Distance (NHD). Results suggest that MAC’s multi-agent collaboration effectively improves LLM-based causal discovery by integrating structured debates and coding execution, with varying performance based on dataset complexity.

**Strengths:**

1. Comprehensive Multi-Agent Approach: The paper introduces a well-structured framework that incorporates debate, planning, and coding agents to address causal inference, allowing MAC to explore causal discovery from multiple angles (e.g., reasoning and statistical analysis).

2. Thorough Evaluation Metrics: The study uses detailed metrics (SHD, NHD) across three datasets with varying complexities, providing an empirical basis for evaluating each model’s effectiveness. This comparison offers valuable insights into MAC’s strengths and areas of improvement in different causal discovery scenarios.

3. Resource Efficiency Analysis: The paper’s analysis of token consumption across models (e.g., Coding-Debating Hybrid and Meta Agents Model) offers practical insights into optimizing multi-agent frameworks, allowing future research to refine token efficiency in LLM-based causal discovery.

**Weaknesses:**

1. Lack of Formal Framework for Agentic Workflow Convergence: Although MAC leverages multi-agent interactions for causal discovery, it lacks a formal mathematical model that defines or ensures convergence of the agents' outputs to a stable causal graph. In high-dimensional causal graphs, where causal inference is combinatorially complex, the absence of convergence guarantees can lead to oscillating or suboptimal solutions.

2. Inadequate Quantification of Noise Propagation in Causal Inference: While the study qualitatively discusses errors in causal relationships and agentic noise, it lacks a quantitative model of error propagation, particularly in the Coding Agents and Hybrid Models. In causal discovery, noise in initial data or agentic decision-making can propagate through the inference process, leading to accumulated inaccuracies.

3. Limited Theoretical Justification for Agent Roles and Workflow Sequencing: The paper introduces different agent roles (debate agents, coding agents) and workflows (Meta, Coding, and Hybrid Models) but lacks a theoretical basis for why certain roles or workflows are expected to perform better in specific causal inference tasks. For example, why should debates be prioritized in certain instances over coding, or vice versa? Without a rigorous framework, the choice of agent configurations appears ad hoc.

4. Scalability Constraints in Token and Computational Costs: The MAC framework requires significant token and computational resources, especially in complex datasets or with high-dimensional graphs. Although the paper provides empirical token usage data, it does not propose concrete solutions for reducing token or computational overhead, limiting MAC's practical applicability to smaller datasets or high-resource settings.

Summary: While MAC offers a novel multi-agent approach for LLM-based causal discovery, its theoretical framework and practical scalability are limited. The lack of formal convergence guarantees, quantification of noise propagation, and rigorous justification for agent workflows restrict its contribution to the field. Future work that addresses these theoretical gaps and practical constraints could make MAC a more impactful framework for causal discovery with LLMs.

**Questions:**

1. Generalizability Across Different Datasets: How would MAC’s performance vary on datasets with more complex causal structures or differing distributions? Would integrating domain-specific statistical models or causal priors into the MAC framework improve its adaptability?

2. Impact of Noise Reduction Techniques: Since causal discovery is sensitive to noise, could pre-processing techniques (e.g., noise reduction or causal regularization) improve MAC’s stability in identifying causal graphs? How would the SHD and NHD scores change under varying noise levels?

3. Effectiveness of Hybrid Model Configurations: Given that the Hybrid Model combines Meta-Debate and Coding Agents, what is the optimal sequencing or balance between these agents for tasks with different complexities? Would tasks benefit from a dynamic hybrid approach that adjusts the agent workflow based on intermediate inference outcomes?

4. Convergence Criteria and Agentic Redundancy: Does the MAC framework have a convergence criterion to stop debate or coding cycles, especially when agents are unlikely to yield new information? What role could redundancy elimination techniques play in enhancing efficiency, and are there scenarios where additional agents create diminishing returns?

---

### Official Review · Reviewer_Rq6R · 2024-11-03

**Soundness:** 1
**Presentation:** 1
**Contribution:** 2
**Rating:** 1
**Confidence:** 3

**Summary:**

This paper proposed a meta-debator model for causal discovery using large language models called MAC and perform simulations on three datasets.

**Strengths:**

The idea of using multi-agent systems to solve causal discovery problem is interesting.

**Weaknesses:**

### General

I found it a bit hard to assess the significance of this work, partly because the writing (especially on the experimental setup) is a bit opaque, and I did not find the result to be super exciting. The constant stylish and grammatical imperfections also seem to suggest this paper was written in a rush. I would also recommended the authors proofread the paper and make the writing clearer. Furthermore, mechanically, I do not see how using a multi-agent setup is fundemantally better. I acknowledge that the SHD/NHD metrics were better on some models in authors' experiments, but I am not sure if (1) those improvements were large enough to be significnat; and (2) the inference cost of having more LLMs in the pipeline. I am hoping to see more supporting eveidence that the proposed framework compares favorably to existing methods.




### Writing issues (excerpt)

1. Citation missing at line 150.
2. Empty citations at line 130.
3. Double quotes styld at line 146.
4. Missing period at line 371.
5. Missing figure reference at line 463.
6. Missing figure reference at line 479.
7. Missing period at line 484.
8. Missing period at footnote 2.

**Questions:**

1. Why the Gemini 2 9b rows are empty in Table 2?

2. The results in Table 2 look similar; are bold results significnatly better?

3. How large are the dataset used?

4. What does it mean by "agentic ability" in table 1? What is "multi-agent" in table 1 refers to, does it simply means multiple components are used in MAC?

---

### Note · Authors · 2024-11-26

I have read and agree with the venue's withdrawal policy on behalf of myself and my co-authors.